# Genetic and Epigenetic Biomarkers Associated with Early Relapse in Pediatric Acute Lymphoblastic Leukemia: A Focused Bioinformatics Study on DNA-Repair Genes

**DOI:** 10.3390/biomedicines12081766

**Published:** 2024-08-05

**Authors:** Walaa F. Albaqami, Ali A. Alshamrani, Ali A. Almubarak, Faris E. Alotaibi, Basil Jamal Alotaibi, Abdulrahman M. Alanazi, Moureq R. Alotaibi, Ali Alhoshani, Homood M. As Sobeai

**Affiliations:** 1Department of Science, Prince Sultan Military College of Health Sciences, Dhahran 31932, Saudi Arabia; walbogomi@psmchs.edu.sa; 2Department of Pharmacology and Toxicology, College of Pharmacy, King Saud University, Riyadh 11451, Saudi Arabia; aliadel.almubarak1@outlook.com (A.A.A.); faris.rouqi@gmail.com (F.E.A.); basiljamal.ot@gmail.com (B.J.A.); abdulrahmanmubarak5@gmail.com (A.M.A.); mralotaibi@ksu.edu.sa (M.R.A.); ahoshani@ksu.edu.sa (A.A.); 3Pharmaceutical Care Division, King Faisal Specialist Hospital & Research Centre, Madinah 42523, Saudi Arabia

**Keywords:** acute lymphoblastic leukemia, childhood ALL, precursor-B-ALL, early relapse, late relapse, DNA repair, *PARP1*, miRNA, prognostic biomarker

## Abstract

Genomic instability is one of the main drivers of tumorigenesis and the development of hematological malignancies. Cancer cells can remedy chemotherapeutic-induced DNA damage by upregulating DNA-repair genes and ultimately inducing therapy resistance. Nevertheless, the association between the DNA-repair genes, drug resistance, and disease relapse has not been well characterized in acute lymphoblastic leukemia (ALL). This study aimed to explore the role of the DNA-repair machinery and the molecular mechanisms by which it is regulated in early- and late-relapsing pediatric ALL patients. We performed secondary data analysis on the Therapeutically Applicable Research to Generate Effective Treatments (TARGET)—ALL expansion phase II trial of 198 relapsed pediatric precursor B-cell ALL. Comprehensive genetic and epigenetic investigations of 147 DNA-repair genes were conducted in the study. Gene expression was assessed using Microarray and RNA-sequencing platforms. Genomic alternations, methylation status, and miRNA transcriptome were investigated for the candidate DNA-repair genes. We identified three DNA-repair genes, *ALKBH3*, *NHEJ1*, and *PARP1*, that were upregulated in early relapsers compared to late relapsers (*p* < 0.05). Such upregulation at diagnosis was significantly associated with disease-free survival and overall survival in precursor-B-ALL (*p* < 0.05). Moreover, *PARP1* upregulation accompanied a significant downregulation of its targeting miRNA, miR-1301-3*p* (*p* = 0.0152), which was strongly linked with poorer disease-free and overall survivals. Upregulation of DNA-repair genes, *PARP1* in particular, increases the likelihood of early relapse of precursor-B-ALL in children. The observation that *PARP1* was upregulated in early relapsers relative to late relapsers might serve as a valid rationale for proposing alternative treatment approaches, such as using PARP inhibitors with chemotherapy.

## 1. Introduction

Leukemia is a malignant neoplasm of hematopoietic origin, where neoplastic cells replace bone marrow and peripheral blood cells. Despite the variations of leukemia at both the chronic and acute levels and the specific cell type where the disease has originated, acute lymphoblastic leukemia (ALL) is ranked as the most common type in children, accounting for nearly 75% of leukemia cases, with a peak incidence between 1–4 years [1,2,3]. Despite the latest estimates that the 5-years-overall survival rate of pediatric ALL exceeds 90% [4], therapy failure and disease relapse have been reported in 10–15% of cases, making relapsed ALL one of the top causes of death from childhood cancer [5]. DNA-targeting therapeutics have historically been the cornerstone and the first-line therapy to treat pediatric ALL [6]. Mounting evidence suggests that de novo and acquired resistance of ALL cells to chemotherapy is the foundation of such failure [7]. Thus, understanding the underlying factors impacting the response to ALL chemotherapy at the genomic and genetic levels is a significant step in improving patients’ outcomes and minimizing the rate of disease relapse.

The induction of DNA damage in rapidly growing cells is the basis of chemotherapeutics to eradicate cancer cells [8]. However, regardless of the cancer type, cancer cells have proven their ability to ameliorate chemotherapy-induced genotoxic damage by elevating their DNA repair capacity [9,10]. Such an effect is mediated by DNA-repair mechanisms including direct repair (MGMT), base-excision repair (BER), nucleotide-excision repair (NER), mismatch repair (MMR), Homologues recombination (HR), and non-homologous end joining (NHEJ) [11]. Each pathway recognizes and deals with specific types of DNA damage [12,13]. BER, NER, and MMR remediate single-stranded DNA damage or breaks [14], while HR and NHEJ fix double-strand breaks [15]. Nonetheless, these pathways often exhibit functional overlap. For example, NER and HR cooperate to repair inter-strand crosslink damage [16]. Additionally, NER can act as a backup mechanism for BER and MMR when these pathways are unable to remove their specific DNA damage lesions [17]. Despite efforts to investigate the association between DNA-repair gene dysregulation and relapse in ALL, most studies have been pathway-focused [18,19,20,21,22,23,24,25,26]. Comprehensive analyses encompassing a broader spectrum of DNA-repair mechanisms in ALL remain scarce. The DNA repair capacity is governed by the expression level of repair genes. The expression level of these genes can be regulated on multiple levels, including DNA mutations and polymorphisms, epigenetic alterations, or miRNA-mediated post-transcriptional repression [27].

MicroRNAs (miRNAs) are small, single-stranded, highly conserved non-coding molecules that negatively regulate the expression of target genes post-transcriptionally via mRNA degradation or translation suppression based on the base pairing complementarity [28]. Each of the validated 2300 true human mature miRNAs is capable of regulating the expression of up to hundreds of target genes, and multiple miRNAs can regulate one mRNA [29,30]. Several lines of evidence support the crucial role of miRNAs in regulating the cancer cell response to DNA damage [31,32,33]. Yet, to our knowledge, there is little to no information on the role of these DNA-repair-regulating miRNAs as potential prognostic factors in childhood ALL resistance or relapse.

In this study, we screened for dysregulated DNA-repair genes and their genetic and epigenetic alterations that may be critical in early recurrence (relapse ≤ 2 years) in pediatric ALL. The identified genes and their epigenetic regulators can potentially be used as prognostic biomarkers and a rationale for alternative therapeutic strategies.

## 2. Materials and Methods

### 2.1. Study Design and Patient Selection

The Therapeutically Applicable Research to Generate Effective Treatments (TARGET)—ALL expansion phase II trial was utilized to generate the data for our study [34]. The clinical and genomic profiles of 198 relapsed pediatric precursor B-cell ALL obtained at the time of diagnosis were analyzed using the publicly available site cBioProtal (https://www.cbioportal.org/, 7 March 2022) [35,36]. Patients were stratified into two groups based on their time of relapse: the early-relapsing cohort (*n* = 109), who experienced recurrence within the first two years after remission, and the late-relapsing group (*n* = 89), who relapsed after two years of being declared disease-free.

### 2.2. Clinical Characteristics Evaluation

Clinical data, including age, vital status, overall survival (OS), sex, race, white blood cell count (WBC) at relapse, and genetic abnormalities (TCF3-PBX1, Trisomy 4_10, BCR-ABL1, ETV6-RUNX1, and TP53/ATM mutation status) were compared between early- and late-relapsed cohorts. Continuous variables such as age and WBC were presented as mean ± standard error (SE) and evaluated statistically using Student’s *t*-test. Categorical variables such as vital status, sex, race, and genetic abnormalities were expressed as frequencies and percentages and assessed statically using a Chi-square or Fisher’s Exact Test as appropriate. Kaplan–Meier survival analysis and log–rank tests were computed to analyze OS. The differences between the two cohorts were considered statistically significant if *p* < 0.05. Detailed clinical characteristics are available in Appendix A.

### 2.3. Genetic and Epigenetic Analyses

Comprehensive genetic and epigenetic investigations of 147 DNA-repair genes were conducted in the study [37,38]. Gene symbol, NCBI Entrez, locus, function, and pathway are illustrated in Appendix A. Microarray and RNA-sequencing data have been used to investigate gene expression of the DNA-repair genes independently. Student’s *t*-test was utilized to identify significantly differentially expressed (DE) DNA-repair genes between early- and late-relapse groups. Statistical significance was set at *p* < 0.05. The upregulated genes found in early-relapse groups relative to the late-relapse group at the time of diagnosis and confirmed in both technologies were selected as candidate (hit) genes for further genetic and epigenetic investigations. Venny tool (version 2.1) [39] has been used to identify the DE candidate DNA-repair genes in both RNA-sequencing and Microarray platforms.

To study the underlying factors for the upregulation of the candidate genes, genetic alternations (somatic mutations and putative copy number alterations [PCNA]) and epigenetic dysregulation (gene methylation) were investigated. The assessed Somatic mutations include missense, inframe (insertion and deletion), and truncating (nonsense, frameshift, non-start, nonstop, and splice). Both amplifications and deletions were examined as PCNA. Log_2_ ratio was calculated as the percentage of early-relapse patients to late-relapse patients who exhibited genetic alterations in the candidate genes. Statistical significance was tested using a two-sided Fisher Exact test, with a significance threshold of *p* < 0.05. For gene methylation, the mean methylation of the candidate genes in each cohort was computed, and then the log_2_ ratio was obtained. Student’s *t*-test was used to assess the statistical significance.

### 2.4. miRNA Signature and Target Genes

Differentially expressed miRNAs between early and late relapsers were investigated. A Student’s *t*-test was used to examine the statistical significance of the results. The downregulated miRNAs that might explain the upregulation in the candidate genes were selected for further analyses. Predicted target genes were identified using miRWalk database 3 [40]. Complementary sequences for the seed region in 3′ UTR with scores above 0.95 were considered. The Venny tool (version 2.1) was used to identify the shared miRNAs between the significantly downregulated miRNAs and those found to potentially interact with the candidate genes. These shared miRNAs were validated using miRTarBase 9.0 [41], an experimentally validated miRNA-gene interactions database.

### 2.5. Survival Analyses

Kaplan–Meier free and overall survival analyses were conducted to evaluate the impact of candidate genes on the patients’ overall and free survival. The mean was chosen as a cutoff value to categorize the patients into two groups: patients who gained high-expression values of a given gene (>mean value) and patients who gained low-expression values (<mean value). Kaplan–Meier curves were generated based on single-gene expression or combined expression of the candidate genes for both microarray and RNA-sequencing data using GraphPad Prism 9.1 software. Hazard ratios (HRs) and *p* values of the Gehan–Breslow–Wilcoxon test were calculated to estimate the statistical significance differences between high- and low-expression groups.

## 3. Results

### 3.1. Clinical Characteristics of the Enrolled Subjects

Clinical data from 198 pediatric patients were retrieved and analyzed (Table 1). Based on the disease relapse after remission, 109 patients (55.05%) were identified as early relapsers and 89 patients as late relapsers, with a mean age of 9.55 and 7.64 years, respectively (Table 1). Early relapse was associated with higher mortality rates compared with the late-relapse group (77.98% vs. 53.30%; *p* = 0.002), with overall survival (OS) of 23.87 and 76.37 months, respectively (*p* < 0.0001), as shown in Appendix A. The t(1;19)/TCF3-PBX1 translocation has been associated with a poor clinical outcome in childhood ALL [42]. Our results show a significant number of early-relapsed patients were TCF3-PBX1-positive when compared with the late-relapsed group (15 (13.76%) vs. 3 (3.37%); *p* < 0.05), respectively. Patients who relapsed within two years had numerically higher pretreatment white-blood-cell (WBC) counts at relapse diagnosis than patients diagnosed after 2 years of remission (103.47 ± 18.28 vs. 67.92 ± 12.48). There were no significant differences between the two groups in TP53- and ATM-mutation counts. Further details of clinical data are provided in Appendix A.

### 3.2. Differentially Expressed DNA-Repair Genes of Early-Relapsed Patients Relative to Late-Relapsed Patients

Differentially expressed genes were identified between early- and late-relapsed ALL groups using data extracted from TARGET—ALL expansion phase II trial at the cBioPortal Platform (Appendix A). Comprehensive Analysis of Microarray and RNA-sequencing datasets of 147 DNA-repair genes identified 18 and 7 significantly upregulated DNA-repair genes, respectively, in the early-relapse group compared with the late-relapse group (*p* < 0.05) (Table 2 and Table 3). Three genes, namely *ALKBH3*, *NHEJ1*, and *PARP1*, emerged as the differentially expressed genes to be identified by the Microarray and validated by the RNA sequencing (Figure 1). Therefore, these three genes, as potential prognostic factors in early relapse of childhood, were subjected to further investigation.

### 3.3. Genetic Alternations and Methylation Status of the Identified Candidate Genes

To investigate the underlying mechanism resulting in such an overexpression of the three candidate genes in these groups at the time of relapse diagnosis, we performed further subset analyses at the genetic (putative copy number alterations (PCNA) and somatic mutations) and epigenetic levels (gene methylation status and miRNAs expression). No significant abnormal PCNA nor somatic mutations emerged in the three genes in the early-relapse group relative to the late-relapse group (Appendix A). Furthermore, ALKBH3 and PARP1 were hypomethylated in the early-relapse group compared with the late-relapse group (Appendix A). However, such hypomethylation did not reach statistical significance (*p* > 0.05). Interestingly, there were no significant differences between early- and late-relapsing patients in *TP53* and *ATM* mutation counts. In addition, neither gene was significantly upregulated in both the microarray and RNA-sequencing data. These data indicate that the increase in the expression of the three genes is TP53/ATM-independent, and other epigenetic mechanisms could have accounted for the upregulation of these DNA-repair genes in the early relapsers.

### 3.4. miRNA Dysregulation in the Cohort Groups

To gain an insight into the potential role of miRNAs in the significant upregulation of the three candidate DNA-repair genes in early-relapsed patients, we assessed the expression profiles of early and late relapsers from the cBioPortal TARGET trial. We found 93 significantly downregulated miRNAs in early-relapsed patients compared to the late-relapsed group (Appendix A). We then screened for putative binding sites in all known human mRNAs that bind the three candidate DNA-repair genes using the miRWalk database (v3). Of the 93 identified cBioPortal miRNAs, we identified 45 miRNAs that predictively bind the 3’ UTR of these DNA-repair genes. Among the three candidate DNA-repair genes, *NHEJ1* emerged as the top gene to theoretically be regulated by 44/45 (97.77%) of the identified miRNAs, whereas *ALKBH3* and *PARP1* were regulated by four and three miRNAs, respectively (Appendix A). Interestingly, upon filtration of the predicted gene-targeting miRNAs based on literature validation using the miRTarBase database, we identified hsa-miR-1301-3p as a validated miRNA to target *PARP1*.

### 3.5. Stratification of Clinical Outcome Based on the DNA-Repair Gene Expressions at Diagnosis

To assess the correlation between the overexpressed DNA-repair candidate genes with the relapse-free survival (RFS), we stratified the patients into high- and low-expression subgroups of each of the three genes at diagnosis to create Kaplan–Meier (KM) plots. In the Microarray dataset, there was a significant correlation between the expression of each of the three candidate DNA-repair genes and the length of the RFS (Figure 2). The median RFS time was 15.65 months for *PARP1*, 17.60 months for *NHEJ1*, and 18.27 months for *ALKBH3* high-score group vs. 26.83 months, 27.25 months, 26.13 months for low-expression groups, respectively. In the combined dataset, as expected, the median RFS time was 26.13 months for the low-expression score cohort (*n* = 101) and 16.33 months for the high-expression score cohort (*n* = 66). The log–rank hazard ratio for the high-genes score cohort relative to the low-genes score cohort was 1.653 (95% CI = 1.180 to 2.314).

In the RNA-sequencing dataset (Figure 3), only the high-score *ALKBH3* cohort was significantly correlated with shorter RFS with a median time of 21.72 months compared with 26.70 months in the low-score group [*p* = 0.021; HR 1.550 (95% CI = 1.050 to 2.289)]. Both stratified *NHEJ1* and *PARP1* groups failed to significantly correlate with a shorter median RFS time (*p* = 0.129 and *p* = 0.534), respectively. Nevertheless, combining the datasets of the three genes revealed a significantly shorter median RFS time (29.40 months) in the high-expression group (*n* = 50) compared to 40.60 months for the low-expression group (*n* = 50) (*p* < 0.05). These data suggest that pediatric ALL patients with higher expression of DNA-repair genes are more likely to relapse earlier than those with lower expression, further substantiating their role as potential prognostic biomarkers.

### 3.6. Expression Levels of miR-1301-3p Correlate with Poorer Patient Outcomes

We performed KM survival analysis to obtain insights about the clinical implications of *PARP1*-targeting miR-1301-3*p* in pediatric ALL patients. miR-1301-3*p* was significantly downregulated in early relapsers compared with late ones (Figure 4A). The predicted consequential pairing of *PARP1* with miR-1301-3*p* is shown in Figure 4B. As demonstrated in Figure 4C, the low expression of miR-1301-3*p* was numerically associated with shorter median disease-free survival at 31.50 months vs. 66.80 months of the higher-expression group (*p* = 0.086). However, the median overall survival time was significantly shorter in patients with lower miR-1301-3p expression levels (*p* = 0.005) (Figure 4C). Such observation could indicate the molecular mechanism by which *PARP1* was upregulated in early relapsers.

## 4. Discussion

Resistance to chemotherapy is classified into intrinsic and acquired, depending on the resistance timing, pretreatment or posttreatment [44,45]. Distinctive mechanisms have been demonstrated to drive the development of the two resistance categories [45]. Enhanced DNA-repair is one of the unique features by which tumor cells escape death induced by DNA-damaging chemotherapeutics [46]. Such an innate fortification against cytotoxic insults could develop despite the presence of chemotherapeutic treatment, increasing the chances of developing early relapse to these therapies [13,47]. This insinuates the need for new ways to predict relapse so that appropriate treatment interventions can be made early to improve patients’ outcomes and minimize the rate of disease relapse. In this study, we sought to identify gene-expression signatures that distinguish early-pediatric ALL relapsers from late relapsers at the time of diagnosis so they can be used as predictors of relapse and rationale for alternative therapeutic plans. We further investigated the genetic and epigenetic mechanisms underlying the differentially expressed genes among the two cohorts. Microarray and RNA-sequencing datasets confirmed three DNA-repair genes, *ALKBH3*, *NHEJ1*, and *PARP1*, to be significantly upregulated in the early relapsers compared with late relapsers. These results suggest that three genes might contribute to the intrinsic resistance that the early-relapsing cohort exhibited, which may have led to the failure of therapeutic regimes. Furthermore, our data suggest that miRNA-driven epigenetic alterations were primarily responsible for the resulting overexpression of the three DNA-repair genes and, thus, the acquisition of the initial drug resistance in the early relapsers.

*ALKBH3* belongs to a family of DNA dealkylating enzymes that remedy the single-stranded DNA (ssDNA) damage by the BER mechanism [48,49]. Such a repair mechanism aims to correct DNA strand breaks, including single-base oxidation, DNA adducts, and inter- and intra-strand covalent crosslinks caused by DNA-damaging chemotherapeutic agents [50]. *ALKBH3* upregulation has been associated with poorer prognostic outcomes in several cancers, including non-small-cell lung cancer [51], pancreatic cancer [52], renal cell carcinoma [53], and hepatocellular carcinoma [54]. In accordance with those studies, our study is the first to demonstrate, in early-pediatric ALL relapsers, a significant correlation between *ALKBH3* overexpression and shorter free survival. These findings reveal the essential role of *ALKBH3* in chemoresistance to certain genotoxic reagents and, thus, disease relapse.

DNA double-strand breaks (DSBs) are the most significant lesions, which are mainly repaired by NHEJ and HR pathways [55]. Mechanistically, unlike HR, which uses a homologous DNA sequence as a template to guide the error-free restoration of broken DNA, the NHEJ pathway directs re-ligation of the broken DNA molecule in a template-independent manner [56]. Dysregulated NHEJ expression has been associated with several cancers’ development, progression, chemoresistance, and relapse, including hematological malignancies [57]. Findings from the study by Chiou et al. [58] revealed that ALL patients displayed elevated levels of NHEJ factors prior to treatment compared to controls, which were then downregulated after patients achieved complete remission. Interestingly, the downregulated NHEJ was then overexpressed only in relapsed cases. In another study, lower expression of NHEJ factors was associated with better ALL disease remission and more favorable clinical outcomes than patients with high expression [59]. In accordance with the existing literature, our results show that pediatric ALL patients with higher expression of the *NHEJ1* gene are more likely to relapse earlier than those with lower expression. Despite insignificant RNA-sequencing data differences, Microarray data analysis indicated that higher *NHEJ1* expression was strongly linked with shorter survival times, suggesting its potential usefulness as a prognostic biomarker. However, the absence of existing inhibitors of both *ALKBH3* and *NHEJ1* undermines their importance as a therapeutic strategy to prevent or minimize the chance of ALL disease relapse after complete remission. Our findings suggest that it would be key to future studies to investigate the impact of inhibiting those genes by using inhibitor molecules, siRNAs, or other gene-editing techniques on the timing of disease relapse and patients’ survival.

*PARP1* is a member of a large family of 17 Poly (ADP-ribose) polymerases involved in several cellular processes, including chromatin remodeling, stress response, and ssDNA repair. The role of *PARP1* has been demonstrated to extend beyond ssDNA repair to serve as a pivotal player in the action of other DNA-repair mechanisms, including NER, MMR, HR, and NHEJ [60]. *PARP1* upregulation has been reported in several malignancies, including oral cancer [61], ovarian cancer [62], testicular tumors [63], neuroblastoma [64], malignant lymphoma [65], breast cancer [66,67], colon cancer [68], endometrial cancer [69], and BRCA-mutated ovarian cancer [70,71]. In acute myeloid leukemia (AML), patients with high *PARP1* expression had significantly shorter overall survival and free survival times compared with the low-expression group [72]. Yet, the role of *PARP1* in pediatric ALL relapse remains elusive. Similar to previously published studies, we report that higher expression of *PARP1* was strongly associated with early ALL relapse and shorter survival times compared with the low-expression cohort.

We analyzed the miRNA expression profiles of early and late relapsers to understand the potential mechanism by which *PARP1* was upregulated in early relapsers. We identified miR-1301-3p as the validated miRNA, possibly targeting *PARP1* at its 3′-UTR [43]. Significant downregulation of miR-1301-3p accompanied *PARP1* upregulation in the early relapsers. Several lines of evidence reported similar downregulation of miR-1301-3p in 11 types of cancer, indicating its action as a tumor suppressor miRNA [73]. Our data show that miR-1301-3p can be an independent prognostic biomarker of pediatric ALL. The observation that early-relapse patients had their *PARP1* inhibitor repressed suggests that using *PARP1* inhibitors could serve as a potential solution to the problem of early relapse of pediatric ALL after patients’ complete remission. It is important to consider other genes that might govern *PARP1* function. For example, *MACROH2A1* was found to crumble PARP1 activity by a direct interaction [74]. *TP53* [75], *KHDRBS1* [76], and *HPF1* [77] also have been found to influence PARP1 function. Thus, future studies are much needed to investigate the role of these genes and other regulatory genes in promoting *PARP1* upregulation in early-relapsing patients.

Unlike the other two DNA-repair genes identified in this study, *PARP1* is the only gene with several FDA-approved inhibitors [78]. These inhibitors have been developed and approved as a holistic approach for maintaining therapy in the recurrence of several cancers, including triple-negative breast cancer, ovarian, gastric, small-cell, and non-small-cell lung cancers [79,80]. There are several ongoing clinical trials on the efficacy of Olaparib, Rucaparib, Niraparib, Veliparib, and, more recently, Talazoparib in overcoming *PARP1*-mediated resistance in hematological malignancies [80]. Even though all these studies focus on AML rather than ALL, the promising results of those trials urge us to investigate the same concept in other types of leukemias. The complementary DNA-damaging effects of PARP inhibitors and chemotherapy have provided a strong rationale for exploring their combination. To our knowledge, our study is the first to show the correlation between *PARP1* upregulation and the development of early relapse in pediatric ALL patients and the first to establish the need for further studies to investigate the impact of concomitant use of *PARP1* inhibitors on the ALL disease recurrence after complete remission.

The existence of a few limitations urges for careful interpretations of our findings. For instance, the expression findings are relative between early- and late-relapsing patients. We cannot extrapolate our findings to cover other cohorts, such as relapse-free patients. Future work, including non-diseased and relapse-free cohorts, is warranted to comprehensively assess the role of DNA-repair mechanisms in ALL progression and treatment. In addition, our study is a secondary analysis of well-established data sets, meaning that all clinical inclusion and exclusion criteria were previously determined. Several uncounted confounders might have impacted the observations reported in this study between the two groups, including which chemotherapeutic drug was used, the number of doses given, the time frame when each treatment was initiated, and the definition of complete remission. Moreover, the existence of other comorbidities or the concurrent use of other treatments are also limitations in the study. However, to our knowledge, this study is the most comprehensive investigation examining the association between DNA-repair genes, genetically and epigenetically, and time of recurrence in such a large relapsing pediatric ALL cohort. The fact that this study proposes the concurrent use of *PARP1* inhibitors with chemotherapy in pediatric patients with ALL based on two independent techniques is a critical finding for this study. Nevertheless, future studies are warranted to validate these findings and to clinically assess the potential use of *PARP1* inhibitors in high-risk patients to ameliorate ALL response to chemotherapy and prevent patients’ relapse.

## 5. Conclusions

The DNA-repair gene expression of relapsing ALL in children showed distinct profiles depending on the timing of relapse. The findings presented in this study were based on the first analysis of one publicly available Microarray dataset and then an independent confirmation by a secondary analysis of the RNA-sequencing dataset. Children who relapsed early (≤24 months) had significantly higher *ALKBH3*, *NHEJ1*, and *PARP1* expression than those who relapsed later (>24 months). The upregulation of *PARP1* was accompanied by a significant downregulation of the validated targeting miRNA, miR-1301-3p, in early relapsers. This might provide a valid rationale for proposing the use of *PARP1* inhibitors as an adjuvant therapy with chemotherapeutics to prevent therapy failure and disease relapse in pediatric ALL.

## Figures and Tables

**Figure 1 biomedicines-12-01766-f001:**
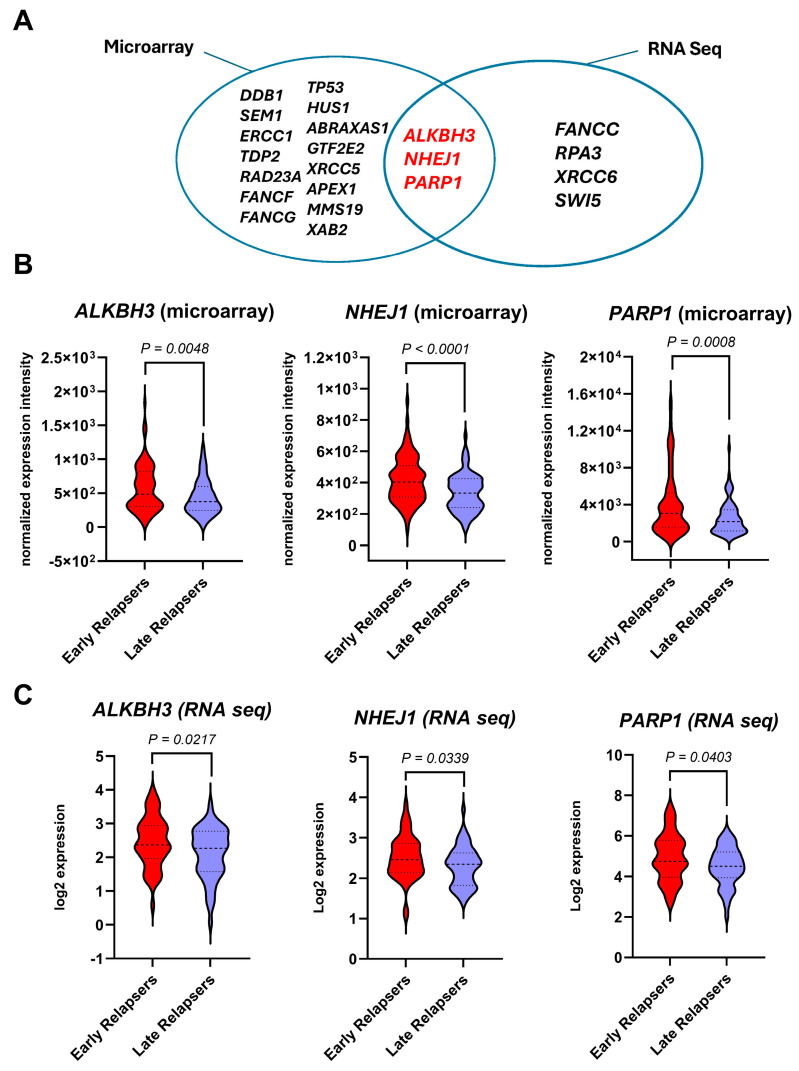
Significantly upregulated DNA-repair genes in early-relapsing patients compared with late-relapsing patients. (**A**) 18 and 7 genes were found overexpressed in the early-relapse group relative to the late-relapse group using microarray and RNA seq, respectively. ALKBH3, NHEJ1, and PARP1 were found upregulated in the early-relapsing group (red) relative to the late-relapsing group (purple) in microarray (**B**) and RNA seq (**C**) datasets. Bold intermittent lines represent the mean values, while light intermittent lines represent 95% confidence interval values. Statistical analysis was computed using Student’s *t*-test.

**Figure 2 biomedicines-12-01766-f002:**
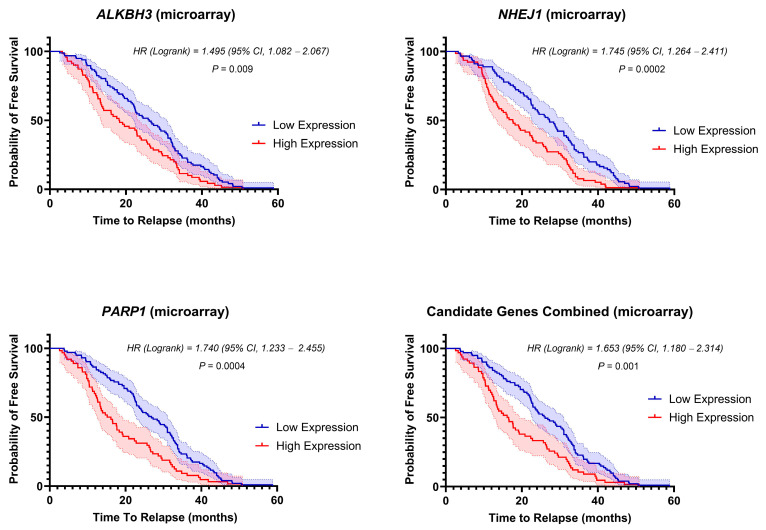
Free survival curves for low-expression (blue) and high-expression patients (red) based on the expression of *ALKBH3*, *NHEJ1*, *PARP1*, and the three candidate genes combined (microarray). The shaded area represents the 95% confidence interval (CI) for each curve. Hazard ratio (HR) and *p* value were calculated using the log–rank test.

**Figure 3 biomedicines-12-01766-f003:**
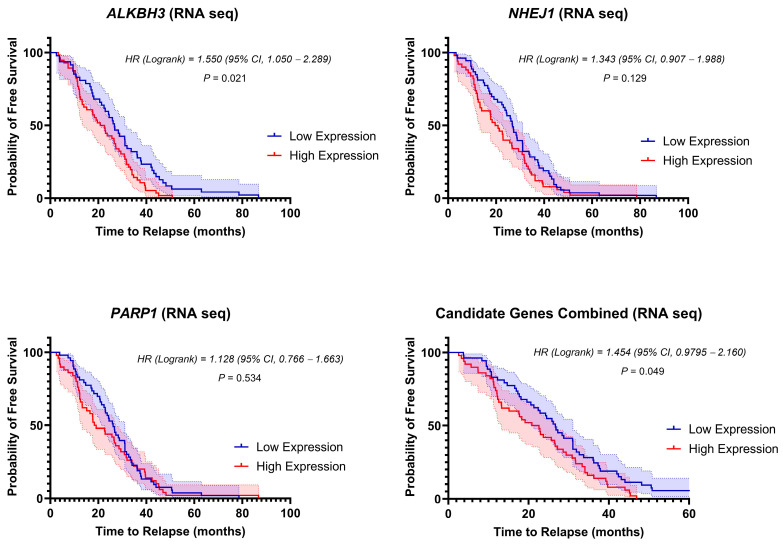
Free survival curves for low-expression (blue) and high-expression patients (red) based on the expression of *ALKBH3*, *NHEJ1*, *PARP1*, and the three candidate genes combined (RNA sequencing). The shaded area represents the 95% confidence interval (CI) for each curve. Hazard ratio (HR) and *p* value were calculated using a log–rank test.

**Figure 4 biomedicines-12-01766-f004:**
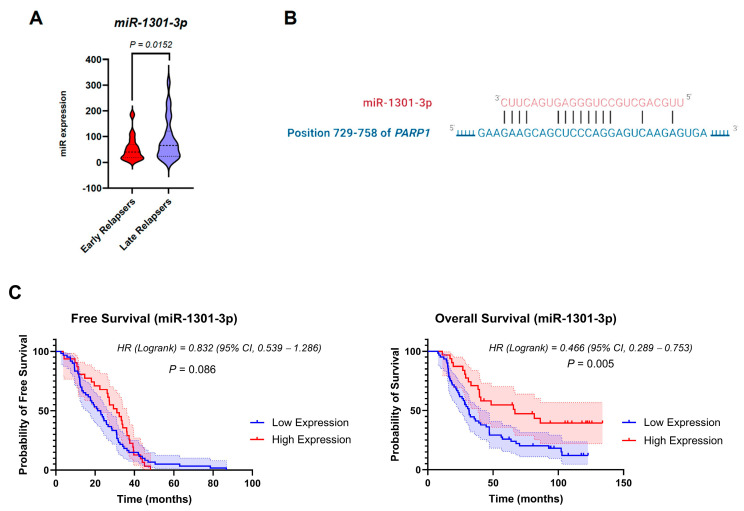
Survival curves for low-score patients (blue) and high-score patients (red) based on the expression of miR-1301-3p. (**A**) miR-1301-3p expression levels among early (red) and late relapsers (purple). Bold intermittent lines represent the mean values, while light intermittent lines represent 95% confidence interval values. Statistical analysis was computed using Student’s *t*-test. (**B**) validated binding sites between PARP1 and miR-1301-3p as identified by [43]. (**C**) The shaded areas represent the 95% confidence interval (CI) for each curve. Hazard ratio (HR) and *p*-value were calculated using a log–rank test.

**Table 1 biomedicines-12-01766-t001:** Clinical characteristics of relapsed ALL patients included in the study.

Parameter	Early-Relapsed *n* = 109 (55.05%)	Late-Relapsed *n* = 89 (44.95%)	*p*-Value *
Age (years)	9.55 ± 0.62	7.64 ± 0.53	0.023
Vital status	Alive = 24 (22.02%) Deceased = 85 (77.98%)	Alive = 38 (42.70%) Deceased = 51 (53.30%)	0.002
Overall Survival (months)	23.87	76.37	<0.0001
Sex	Male = 67 (61.47%) Female = 42 (38.53%)	Male = 45 (50.56%) Female = 44 (49.44%)	0.124
race	White = 73 (66.97%) African American = 14 (12.84%) Others = 3 (2.75%) Unknown = 19 (17.43%)	White = 69 (77.53%) African American = 5 (5.62%) Others = 2 (2.25%) Unknown = 13 (14.61%)	0.294
WBC	103.47 ± 18.28	67.92 ± 12.48	0.127
TCF3-PBX1 fusion status	Positive = 15 (13.76%) Negative = 60 (55.05%) Unknown = 34 (31.19%)	Positive = 3 (3.37%) Negative = 59 (66.29%) Unknown = 27 (30.34%)	0.032
Trisomy 4_10	Positive = 5 (4.59%) Negative = 91 (83.49%) Unknown = 13 (11.93%)	Positive = 7 (7.86%) Negative = 67 (75.28%) Unknown = 15 (16.85%)	0.356
BCR-ABL1 fusion status	Positive = 2 (1.83%) Negative = 106 (97.25%) Unknown = 1 (0.92%)	Positive = 89 (100%) Negative = 0 (0%) Unknown = 0 (0%)	0.288
ETV6-RUNX1 fusion status	Positive = 5 (4.59%) Negative = 88 (80.73%) Unknown = 16 (14.68%)	Positive = 11 (12.36%) Negative = 64 (71.91%) Unknown = 14 (15.73%)	0.123
TP53 mutation status	Positive = 3 (2.75%) Negative = 62 (56.88%) Unknown = 44 (40.37%)	Positive = 0 (0%) Negative = 65 (62.92%) Unknown = 33 (37.08%)	0.156
ATM mutation status	Positive = 3 (2.75%) Negative = 62 (56.88%) Unknown = 44 (40.37%)	Positive = 0 (0%) Negative = 65 (62.92%) Unknown = 33 (37.08%)	0.156

* *p*-value based on Chi-square test, Fisher-exact test, or Student’s *t*-test as appropriate.

**Table 2 biomedicines-12-01766-t002:** Significantly upregulated DNA-repair genes in early-relapsed patients identified by Microarray dataset.

Genes	Fold-Change	*p*-Value *	Pathway
*PARP1*	1.56	7.57 × 10^−4^	Poly(ADP-ribose) polymerase (PARP) enzymes
*TP53*	1.49	8.67 × 10^−4^	Regulation of the cell cycle
*SEM1*	1.31	8.28 × 10^−4^	Homologous recombination
*ERCC1*	1.31	3.60 × 10^−3^	Nucleotide excision repair
*ALKBH3*	1.31	4.78 × 10^−3^	Direct reversal of damage
*DDB1*	1.29	6.80 × 10^−5^	Nucleotide excision repair
*FANCG*	1.29	0.011	Fanconi Anemia
*ABRAXAS1*	1.26	0.0185	Homologous recombination
*NHEJ1*	1.25	6.88 × 10^−5^	Non-homologous end-joining
*RAD23A*	1.24	7.29 × 10^−3^	Nucleotide excision repair
*TDP2*	1.23	4.62 × 10^−3^	Repair of DNA-protein crosslinks
*FANCF*	1.23	9.49 × 10^−3^	Fanconi Anemia
*HUS1*	1.18	7.04 × 10^−3^	Subunits of PCNA-like sensor of damaged DNA
*GTF2E2*	1.18	0.022	Nucleotide excision repair
*XAB2*	1.18	0.0493	Nucleotide excision repair
*APEX1*	1.17	0.0476	Base excision repair
*XRCC5*	1.14	0.0246	Non-homologous end-joining
*MMS19*	1.13	0.0487	Nucleotide excision repair

* *p*-value based on Student’s *t*-test.

**Table 3 biomedicines-12-01766-t003:** Significantly upregulated DNA-repair genes in early-relapsed patients identified by RNA-sequencing dataset.

Genes	Log2 Ratio	*p*-Value *	Pathway
*PARP1*	0.43	0.0403	Poly(ADP-ribose) polymerase (PARP) enzymes
*RPA3*	0.39	0.0217	Nucleotide excision repair
*ALKBH3*	0.34	0.0217	Direct reversal of damage
*SWI5*	0.31	0.048	Homologous recombination
*FANCC*	0.27	2.28 × 10^−3^	Fanconi Anemia
*XRCC6*	0.25	0.023	Non-homologous end-joining
*NHEJ1*	0.22	0.0339	Non-homologous end-joining

* *p*-value based on Student’s *t*-test.

## Data Availability

ALL TARGET-phase II datasets that were investigated are publicly available at cBioportal database (Available online: https://www.cbioportal.org/study/summary?id=all_phase2_target_2018_pub) (accessed on 7 March 2022).

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
