# Peer review of "Genetic and Epigenetic Biomarkers Associated with Early Relapse in Pediatric Acute Lymphoblastic Leukemia: A Focused Bioinformatics Study on DNA-Repair Genes"

_biomedicines, 2024, doi:10.3390/biomedicines12081766_

Round 1

Reviewer 1 Report

Comments and Suggestions for Authors

The objective of this study was to investigate the function of the DNA repair system and the molecular mechanisms that control the abnormal DNA repair genes in pediatric ALL patients who experience relapse early or late in their treatment. The authors conducted a secondary analysis of the Therapeutically Applicable Research to Generate Effective Treatments (TARGET) – ALL expansion phase II trial, which included 198 children with relapsed pediatric precursor B-cell ALL. The study involved thorough genetic and epigenetic analyses of 132 DNA repair genes. Gene expression was evaluated utilizing Microarray and RNA sequencing technologies. The candidate DNA repair genes were analyzed for genomic alterations, methylation status, and miRNA transcriptome.

Following the introduction to the topic, the author presents the used in the study clearly. The results presented in each section is interesting and a valuable addition to the literature. The figures in the paper provide overall understanding of DNA repair system.

In general, the manuscript is well organized and presented in detail about the state of the art techniques. This paper will definitely impact the field of biomedicines. I recommend the paper for publication after addressing couple of my minor comments.

However, the authors made a comprehensive analysis on the topic presented, I think the authors should more emphasize and illustrate their own understanding on DNA repair gene expression of relapsing ALL in children.

The paper is lack of detailed summary of the advantages and disadvantages of the study presented.

Reviewer 2 Report

Comments and Suggestions for Authors

In this article, Albaqami et al. describe the involvement of DNA repair gene expression in the clinical outcome of pediatric acute lymphoblastic leukemia in terms of relapse. The authors demonstrated upregulation of three DNA repair genes (ALKBH3, NHEJ1, and PARP1) in early relapsed versus late relapsed cases.

This study highlights the importance of DNA repair gene analysis in cancer patient stratification and potential access to precision therapies such as PARP inhibitors or other treatments. Characterization of the functionality of the DNA damage response is critical for access to these specific therapeutic regimens. The study of this functionality is well established in the treatment of solid cancers such as breast and ovarian cancer. 

However, the article presents some points for clarification:

1-Molecular cytogenetic characterization of patients for aneuploidy, additional chromosomal aberrations, TP53 and ATM is mandatory to better analyze the relationship between elevated DNA repair genes and treatment response.

2-Table 1 needs to be corrected regarding the BCR-ABL data and the comparison between the two cohorts. There are errors in the % late relapse for BCR-ABL expression and p-value.

3-The involvement of NHEJ and HR mechanisms in DNA repair should be reviewed in the discussion.

4-The introduction and discussion are too long and should be shortened.

Comments on the Quality of English Language

Reducing the size of the article will give it greater conciseness, precision and clarity 

Reviewer 3 Report

Comments and Suggestions for Authors

In this study, the authors investigated the possible role of dysregulated DNA repair genes as players of early recurrence in pediatric ALL patients. They used publicly available microarray and RNA sequencing datasets from the TARGET-ALL expansion phase II trial to identify gene expression signatures that could distinguish early pediatric ALL relapsers from late relapsers at the time of diagnosis. Among repair genes they selected three genes, ALKBH3NHEJ1, and PARP1, which were significantly overexpressed in children who relapsed early (≤ 24 months) than those who relapsed later (> 24 months) by both microarray and RNA sequencing analyses. Further genetic and epigenetic investigations showed that the upregulation of PARP1 was accompanied by a significant downregulation of the validated targeting miRNA, miR-1301-3p, in early relapsers. Thus, the findings of this study suggest that the possible use of PARP1 inhibitors as an adjuvant therapy with chemotherapeutics to prevent therapy failure and disease relapse in pediatric ALL.

Overall, the manuscript is well written, and findings are properly and clearly described and discussed. The achieved results are very interesting and, if confirmed and corroborated by further investigation, could provide an additional therapeutic approach for preventing pediatric ALL patient relapse. However, as claimed by the authors themselves, further studies are warranted to validate these findings and to clinically assess the potential use of PARP1 inhibitors with chemotherapy as alternative approach, because of the use of well-established datasets. To support their results, the authors could confirm them by analyzing further datasets from alternative similar platforms.

Other comments:

The authors should better explain the method to select the genes involved in DNA repair. Currently, many other genes (not included in the list) have been demonstrated to participate to these critical pathways. For instance, among the others, PKM2 may function as a critical modulator of genomic instability in cancer cells; interestingly, a study also reported a role of PKM2 in PARP1 mediated DNA damage repair [reviewed in doi: 10.1016/j.bbcan.2024.189089]. Similarly, PRDM2 or MACROH2A1 are critical modulator of BRCA1-dependent genome maintenance [doi: 10.1016/j.celrep.2014.07.024].

Round 2

Reviewer 2 Report

Comments and Suggestions for Authors

thank you for your update